# Non-Neuronal Acetylcholinesterase Activity Shows Limited Utility for Early Detection of Sepsis

**DOI:** 10.3390/biomedicines11082111

**Published:** 2023-07-26

**Authors:** Aleksandar R. Zivkovic, Karsten Schmidt, Stefan Hofer, Thorsten Brenner, Markus A. Weigand, Sebastian O. Decker

**Affiliations:** 1Department of Anesthesiology, Heidelberg University Hospital, 69120 Heidelberg, Germany; 2Department of Anesthesiology and Intensive Care Medicine, University Hospital Essen, University Duisburg-Essen, 45147 Essen, Germany; 3Clinic for Anesthesiology, Intensive Care, Emergency Medicine I and Pain Therapy, Westpfalz Hospital, 67661 Kaiserslautern, Germany

**Keywords:** cholinesterase, cholinergic anti-inflammatory pathway, extrasynaptic, neuro-immune, inflammatory response, intensive care unit, infection

## Abstract

(1) Background: Sepsis is a severe systemic inflammatory condition characterized by rapid clinical deterioration and organ dysfunction. The cholinergic system has been implicated in modulating the inflammatory response. Acetylcholinesterase (AChE), an enzyme primarily responsible for the hydrolysis of acetylcholine, has been proposed as a potential early indicator of sepsis onset. However, the exact role of non-neuronal AChE activity in sepsis and its correlation with disease severity and patient outcomes remain unclear. This study aimed to investigate the involvement of AChE activity in sepsis and evaluate its association with disease severity and clinical outcomes. (2) Methods: A prospective study included 43 septic patients. AChE activity was measured at sepsis detection, as well as 7 and 28 days later. Inflammatory biomarkers, disease severity scores, and patient outcomes were evaluated. (3) Results: AChE activity remained stable for 7 days and decreased at 28 days. However, there was no correlation between initial AChE activity and inflammatory biomarkers, disease severity scores, ICU stay, or hospital stay. (4) Conclusions: Non-neuronal AChE activity may not reliably indicate early sepsis or predict disease severity.

## 1. Introduction

Systemic inflammation, characterized by an immediate immune response to noxious stimuli, often leads to rapid clinical deterioration and organ dysfunction, necessitating prompt intervention for improved patient outcome [1]. The cholinergic system has been recognized as a modulator of the inflammatory response, with evidence suggesting its inhibitory effect on pro-inflammatory processes and facilitation of anti-inflammatory cytokine release [2,3,4,5]. While direct measurement of cholinergic activity remains challenging, assessing the activity of acetylcholinesterase (AChE), a pivotal enzyme responsible for cholinergic activity, may provide valuable insights.

Previous study of our group has demonstrated that AChE activity increases during sterile traumatic injury, suggesting its potential role in the immune response to tissue damage [6]. However, its behavior during sepsis, another condition associated with systemic inflammation, remains unclear. AChE, primarily found in neuromuscular and cholinergic synapses of the central nervous system, as well as in erythrocytes, is responsible for rapid hydrolysis of synaptic acetylcholine [7,8,9]. Moreover, non-neuronal cholinesterases can modulate the cholinergic anti-inflammatory response by hydrolyzing plasmatic (non-neuronal) acetylcholine [10,11,12,13]. However, the physiological function of extrasynaptic AChE and its involvement in immune responses remain poorly understood.

In this study, we aimed to investigate whether bedside-measured AChE activity in blood could serve as an early indicator of emerging systemic inflammation. Specifically, we focused on patients with sepsis and examined both short-term and long-term effects of AChE activity on disease severity. Additionally, we aimed to assess whether changes in initially measured AChE activity could predict patient outcome following sepsis, considering its pivotal role as an enzyme responsible for modulation of cholinergic activity. This study serves as an exploratory investigation, aiming to provide preliminary insights into the relationship between AChE activity and sepsis. We hypothesize that AChE activity may be altered during sepsis, similar to sterile traumatic injury, and could potentially serve as an early indicator of disease progression.

## 2. Materials and Methods

### 2.1. Study Design

This study represents a secondary analysis of a prospective observational study that was originally designed to detect mycoses in patients with septic shock [14]. The study was conducted at the intensive care unit of Heidelberg University Hospital and was approved by the Ethics Committee of the Medical Faculty at Heidelberg University (File Number: S-097/2013). Informed consent was obtained from all patients or their legal representatives. Patients included in this study were diagnosed with sepsis based on the criteria outlined in the Surviving Sepsis Campaign: International Guidelines for Management of Severe Sepsis and Septic Shock 2012 [15]. Out of the initially recruited 50 patients, 7 individuals were excluded from the study due to missing initial AChE measurements. This resulted in a final study population of 43 patients (Figure 1). Basic demographic data, site of infection, and patient outcomes are presented in Table 1.

### 2.2. Measurements

Blood samples used in this study were collected at the time of sepsis detection, followed by additional blood samples obtained 7 and 28 days later. AChE enzyme activity was measured using the ChE Check point-of-care-testing (POCT) device (Securetec Detektions-Systeme AG, Neubiberg, Germany; In-Vitro-Diagnostics Guideline 98/79/EG; DIN EN ISO 18113-2 and -3) according to the manufacturer’s instructions, as previously described by Zivkovic et al. [16]. Enzyme activity was measured in U/mg Hb (hemoglobin). The analysis of inflammatory biomarkers, including white blood cell count (WBCC) and C-reactive protein (CRP), was conducted following the standardized protocols of the central laboratory of the Heidelberg University Hospital. Disease severity of septic patients was assessed at the intensive care unit using the Acute Physiology and Chronic Health Evaluation (APACHE)-II and Sequential Organ Failure Assessment (SOFA) scores. Patients were routinely scored according to the protocols of the intensive care unit.

### 2.3. Statistical Analysis

Data were compiled in an electronic database using Microsoft Excel for Microsoft 365 MSO (Version 2008, Microsoft Corp., Redmond, WA, USA). GraphPad Prism 9 (GraphPad Software, La Jolla, CA, USA, www.graphpad.com) was employed for data evaluation. The Gaussian distribution of the study groups was verified using the D’Agostino and Pearson omnibus normality tests. The data are presented as median with interquartile range (IQR). Statistical significance between the study groups was determined using the mixed effects analysis or Mann–Whitney test. Correlation analysis was performed using the Spearman correlation test. The best-fit value was calculated using a simple linear regression model. A *p*-value <0.05 was considered statistically significant.

## 3. Results

The study included 43 patients admitted to the intensive care unit with diagnosed sepsis. Patient and disease characteristics are presented in Table 1. Plasma concentrations of inflammatory biomarkers, such as CRP and WBCC, were already significantly increased at the time of admission. Subsequent measurements obtained 7 and 28 days following sepsis onset remained increased (Table 2).

Concurrent measurements of AChE in blood of septic patients showed a sustained enzymatic activity during the initial 7 days following ICU admission. A mild but significant reduction in AChE activity was observed in the later phase: 28 days following admission (*p* = 0.0106, mixed-effects analysis, Table 2, Figure 2). Furthermore, there was no correlation between AChE and the above-mentioned inflammatory biomarkers, when measured at the admission to the ICU (Figure 3).

We next tested whether the activity of AChE obtained from 32 patients surviving a 28-day period following sepsis onset differs from that of 11 patients who did not survive. AChE activity obtained from 28-day survivors was 42 (37–46) U/mg Hb, compared to 41 (36–48) U/mg Hb in non-survivors. In addition, AChE activity obtained from 27 patients surviving 90 days was 43 (37–46) U/mg Hb, compared to 41 (37–48) U/mg Hb measured in 90-day non-survivors, revealing comparable results. No significant difference in AChE activity was observed between patients who survived sepsis and those who did not survive within both the 28-day (*p* = 0.8747; Mann–Whitney test; Figure 4a) and 90-day periods after ICU admission (*p* = 0.8766; Mann–Whitney test; Figure 4b).

Furthermore, we conducted a correlation analysis to investigate the relationship between AChE activity and the length of stay in septic patients. Findings revealed no significant correlation between AChE activity and either the duration of ICU stay or the length of hospital stay (Figure 5). 

Finally, we tested whether AChE activity, obtained at the ICU admission, correlated with the severity of the disease. No significant correlation was observed between the initial AChE activity measurement and concurrently obtained disease severity scores (SOFA and APACHE II) from septic patients (Figure 6).

## 4. Discussion

The findings of our study provide valuable insights into the behavior of acetylcholinesterase activity in sepsis and its relationship with disease progression and outcomes. Contrary to our initial hypothesis, we observed that AChE activity did not correlate with inflammatory biomarkers at the time of sepsis detection. This finding suggests that AChE activity may not serve as an early indicator of pathogen-induced systemic inflammation and sepsis.

One interesting finding of our study is that AChE activity remained unaltered during the first 7 days following sepsis detection. This contrasts with previous studies that demonstrated a rapid modulation of the cholinergic system during inflammation [16,17,18,19]. However, we observed a mild but significant reduction in AChE activity 28 days following sepsis detection. One reason for the observed reduction in the enzymatic activity might be a resolving inflammation. Indeed, the delayed activity change observed in AChE did correspond to the decreased levels of CRP, although no significant correlation was found. The lack of correlation between AChE and inflammation biomarkers, such as CRP and WBCC, may only be partly explained by the different dynamics exhibited by these biomarkers. For instance, in contrast to CRP, which exhibited a significant decrease over the 28-day period while still remaining elevated, WBCC demonstrated a sustained increase at day 28, indicating persistent inflammation. This discrepancy suggests that AChE activity may reflect distinct aspects of the inflammatory response that are not solely captured by the standard laboratory biomarkers. The complex and multifaceted nature of sepsis pathophysiology warrants further investigation to fully understand the underlying mechanisms and potential implications of these findings.

The observed discrepancy in our study compared to previous results may be attributed, in part, to the relatively low sample size. With a limited number of participants, the statistical power to detect subtle changes in AChE activity might have been compromised. Sepsis is a heterogeneous condition with varying degrees of inflammatory response among individuals. Small but meaningful changes in AChE activity may have occurred in some patients, but remained undetected due to the limited sample size. Increasing the sample size in future studies would enhance the sensitivity of our findings and improve the ability to detect significant associations between AChE activity and sepsis outcomes.

Additionally, the pathophysiological differences between sterile and pathogen-induced inflammation might contribute to the discrepancies observed in our study compared to previous reports. Sepsis resulting from pathogen-induced inflammation involves complex interactions between the host’s immune response and the invading microorganisms, which can vary depending on the infecting pathogen [20,21]. In contrast, sterile inflammation, such as in traumatic injury, may trigger a different immune response, possibly resulting in altered cholinergic system modulation [22,23,24]. These differing pathophysiological mechanisms could result in distinct dynamics of AChE activity in different inflammatory insults [25,26]. Importantly, cholinergic activity has been shown to act independently of the underlying cause of inflammation and serves as a fast-acting modulating system with anti-inflammatory properties, as previously described by Tracey et al. [27]. Nevertheless, the involvement of the cholinergic system in the context of sterile inflammation and pathogen-induced sepsis may not be uniform. The cholinergic system plays diverse roles in immune regulation, depending on the specific context and timing. Sterile inflammation and pathogen-induced sepsis might trigger different regulatory pathways, leading to divergent effects on AChE activity. The complexity of these interactions warrants further investigation to elucidate the precise role of the cholinergic system in diverse inflammatory conditions.

Furthermore, the administration of multiple medications to sepsis patients may interfere with the cholinergic system, potentially affecting AChE activity. Various medications, such as anticholinergic agents, are commonly used in critical care settings and may modulate the activity of the cholinergic system [28,29,30]. These drugs can impact AChE activity directly or indirectly, leading to changes that might mask or influence the association between AChE activity and sepsis outcomes. Proper consideration and control for the influence of medications on the cholinergic system are essential in future studies to better understand the role of the cholinergic system in sepsis pathophysiology.

We observed that AChE activity at the time of ICU admission did not show correlation with CRP and WBCC. This finding suggests that the cholinergic system may play a role in specific aspects of the inflammatory response in sepsis that are not solely dependent on non-neuronal acetylcholinesterase activity. Additionally, the absence of correlation between AChE activity and disease severity scores or the length of ICU and hospital stay implies that AChE may not be directly linked to the severity of sepsis or predict the clinical course of the disease.

Moreover, the heterogeneity of sepsis and the presence of confounding factors, such as comorbidities and concomitant medications, might also influence AChE activity. Sepsis represents a complex and multifaceted clinical condition, influenced by various patient-specific factors, which could contribute to the diverse dynamics of AChE activity observed in our study compared to previous reports. The importance of conducting disease-specific investigations is underscored, as variations in underlying pathophysiological mechanisms between different types of inflammatory insults may lead to diverse responses in the cholinergic system. Additionally, our study focused on non-neuronal acetylcholinesterase activity, and it is conceivable that other components of the cholinergic system could play a more significant role in sepsis progression. The contrasting results observed in our study underscore the importance of conducting disease-specific investigations and caution against extrapolating findings from one pathological condition to another.

Our results are surprising as they deviate from previous findings in sterile traumatic injury, where initially measured AChE activity correlated with injury severity and the length of ICU stay of injured patients [6]. This discrepancy suggests that the dynamics of AChE activity may differ between different types of inflammatory insults and highlights the complexity of the cholinergic system in the context of immune responses.

Our findings provide a novel long-term perspective on non-neuronal acetylcholinesterase activity during sepsis, demonstrating a sustained activity over time. This observation is surprising, as it contrasts with the previously reported decrease in butyrylcholinesterase, an enzyme closely associated with AChE and known to play an important role during the initial phase of systemic inflammation and sepsis [2,16,19,31,32,33,34,35]. The sustained increase in AChE activity warrants further investigation to better understand its underlying mechanisms and potential implications in the pathophysiology of sepsis.

One possible explanation for our unexpected findings is the timing of sepsis detection and the initial measurement of AChE activity. It is possible that the time point of sepsis detection does not correspond to the actual onset of the disease. This hypothesis is supported by the fact that the initially measured inflammatory biomarkers demonstrated already elevated levels, suggesting a likely delayed time point of sepsis detection. Future studies with a larger sample size and a more precise determination of the sepsis onset time are needed to validate this observation.

Interestingly, our previous study demonstrated significantly reduced acetylcholinesterase activity in healthy volunteers, compared to that obtained from both injured and septic patients, suggesting that the dynamics of AChE changes may be rapid, reflecting the fast-acting modulatory actions of the cholinergic system during inflammation. However, it is important to note that the current study design was not able to capture the potentially narrow time window of AChE-related changes in the cholinergic system following the onset of sepsis.

Indeed, exploring the potential benefits of integrating various biomarkers to achieve better diagnostic efficacy is of the utmost importance. Investigating the synergistic effect of combining AChE activity with other relevant biomarkers in sepsis research holds promise for enhancing our understanding of the disease pathophysiology [36,37]. Biomarkers such as High Mobility Group Box1 (HMGB-1) and TGF-beta 1 have emerged as promising candidates for further investigation in sepsis pathogenesis. HMGB-1’s role as a potent pro-inflammatory cytokine and its association with poor clinical outcomes make it an important target for further study [38,39,40]. TGF beta 1′s complex dual role in sepsis, exhibiting both anti-inflammatory and pro-fibrotic effects, also warrants investigation to understand its contribution to disease progression and potential therapeutic implications [41,42]. Moreover, combining various diagnostic approaches, such as tissue inhibitor of metalloproteinases-2 (TIMP-2), insulin-like growth factor–binding protein 7 (IGFBP7) and free light chains has the potential to accelerate diagnostics and enhance our understanding of the specific changes and impairments occurring in organs during sepsis [43,44,45,46,47]. Large-scale, standardized studies are essential to validate the interactions and patterns among multiple biomarkers, ultimately leading to more personalized diagnostic and therapeutic approaches in sepsis management.

There are several limitations to our study. First, the sample size was relatively small, which may have limited the statistical power to detect subtle changes in AChE activity. Second, the inability to determine the exact time point of sepsis onset limits the generalizability of our findings. Additionally, the heterogeneous nature of sepsis and the presence of confounding factors, such as comorbidities and concomitant medications, could have influenced the observed AChE activity. Future research with larger cohorts and more precise timing of sepsis onset is warranted to further investigate the role of AChE activity in sepsis pathophysiology.

## 5. Conclusions

In conclusion, our study suggests that the AChE assay may not be a suitable method for detecting the onset of sepsis. The limited correlation with disease severity and patient outcomes indicates the need for alternative markers or approaches to assess the involvement of the cholinergic system in sepsis. Further investigations are required to elucidate the complex dynamics of AChE activity and its implications in sepsis.

## Figures and Tables

**Figure 1 biomedicines-11-02111-f001:**
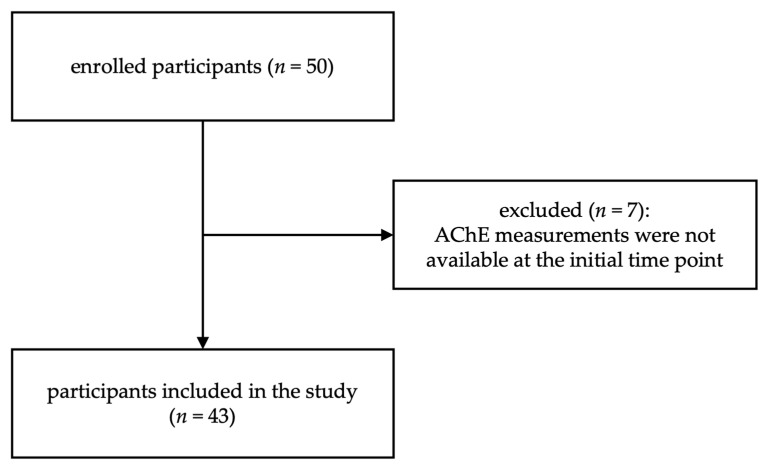
Flow chart representing the recruitment of the study population.

**Figure 2 biomedicines-11-02111-f002:**
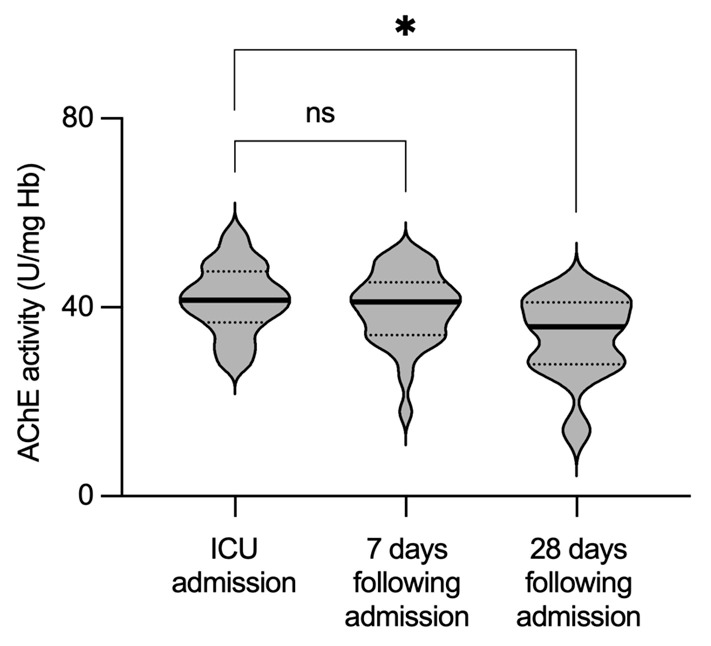
Enzymatic activity of AChE remains unaltered during sepsis. Violin plots represent AChE activity obtained from blood samples of patients with sepsis. Measurements were obtained at the time point of sepsis onset and repeated 7 and 28 days later. Thick black line indicates median values. Dotted lines represent quartiles. AChE—acetylcholinesterase; Hb—hemoglobin; ns—not significant; * *p* < 0.05.

**Figure 3 biomedicines-11-02111-f003:**
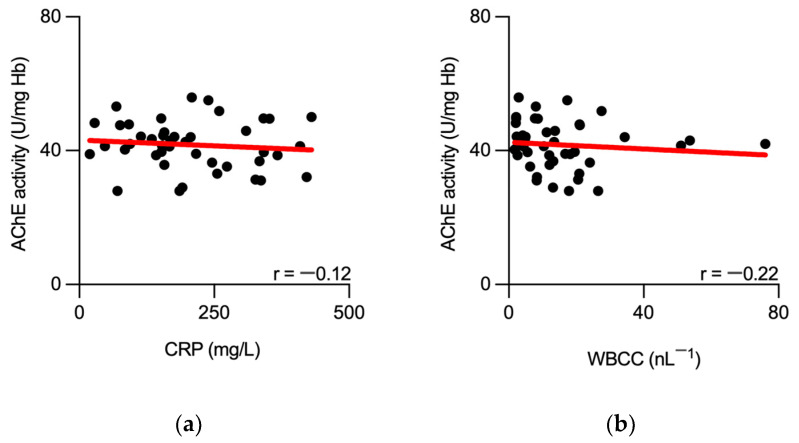
No correlation between initially measured AChE activity and inflammatory biomarkers in septic patients. Scatter plots show no correlation between AChE activity, measured at the sepsis onset, and concurrently measured CRP (**a**) and WBCC (**b**) of septic patients. Black dots represent individual measurements. Best-fit linear regression line is presented in red. AChE—acetylcholinesterase; Hb—hemoglobin; CRP—C-reactive protein; WBCC—white blood cell count; r—Spearman’s correlation coefficient.

**Figure 4 biomedicines-11-02111-f004:**
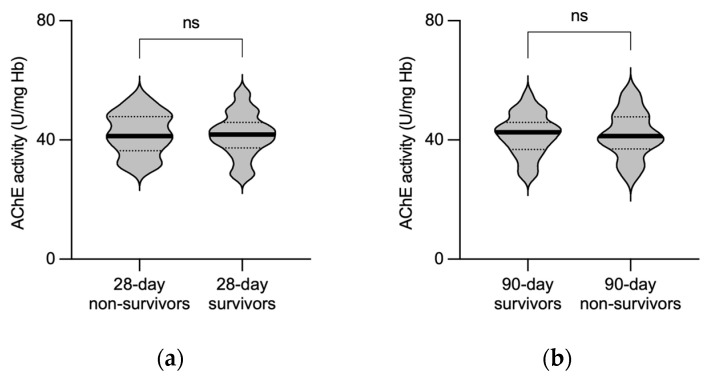
No difference in AChE activity between sepsis surviving and non-surviving patients. Violin plots show enzymatic activity of AChE, measured at the timepoint of sepsis onset from patients who survived and from those who did not survive 28 (**a**) and 90 days (**b**). Thick black line represents median values. Dotted lines represent quartiles. AChE—acetylcholinesterase; Hb—hemoglobin; ns—not significant.

**Figure 5 biomedicines-11-02111-f005:**
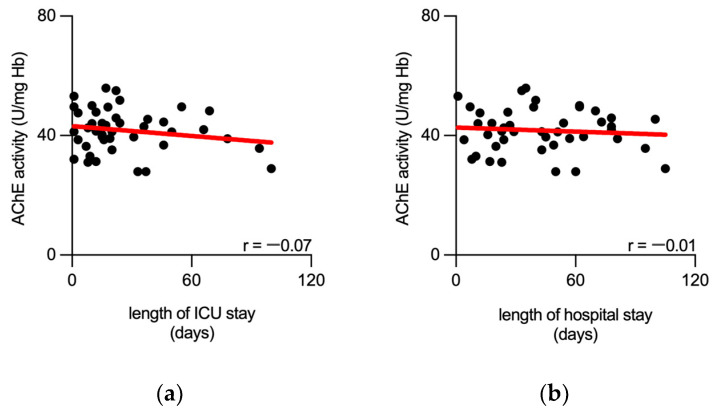
AChE activity measured from patients at sepsis onset does not correlate with the length of stay. Scatter plots show no correlation between AChE activity, measured at the sepsis onset, and length of ICU stay (**a**) nor with the length of hospital stay (**b**) of septic patients. Black dots represent individual measurements. Best-fit linear regression line is presented in red. AChE—acetylcholinesterase; Hb—hemoglobin; r—Spearman’s correlation coefficient.

**Figure 6 biomedicines-11-02111-f006:**
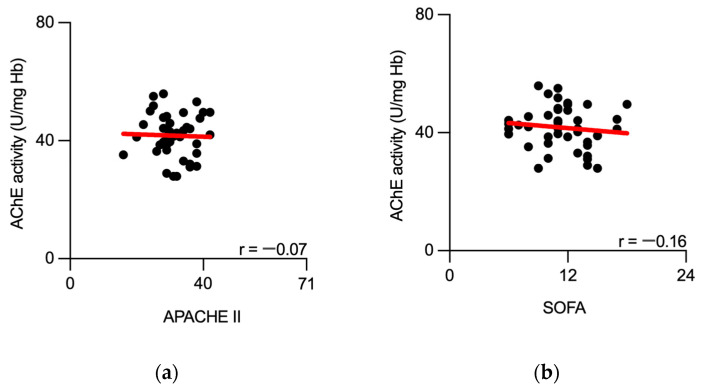
No correlation between initially measured AChE activity and disease severity scores obtained from septic patients. Scatter plots show no correlation between AChE activity, measured at the sepsis onset, and APACHE II (**a**) and SOFA (**b**) scores obtained from septic patients at the hospital admission. Black dots represent individual measurements. Best-fit linear regression line is presented in red. AChE—acetylcholinesterase; Hb—hemoglobin; r—Spearman’s correlation coefficient.

**Table 1 biomedicines-11-02111-t001:** Patient demographics and clinical data.

**Patient data**	
Number of patients	43
Age (years)	66 (59–76) *
Gender (male/female)	31/12
**Septic focus**	
Gastrointestinal tract	35 (81%)
Lung	6 (14%)
Other	2 (5%)
**Length of stay**	
Days in ICU	18 (10–37) *
Days in the hospital	40 (20–62) *

* Median with quartiles.

**Table 2 biomedicines-11-02111-t002:** Summary of the study results.

	ICU Admission	7 Days Following ICU Admission	28 Days Following ICU Admission
AchE (U/mg Hb)	42 (37–48)	41 (34–45)	36 (28–41)
CRP (mg/L)	186 (134–309)	140 (116–181)	70 (63–147)
WBCC (nL^−1^)	12 (5–21)	15 (11–22)	16 (11–23)
SOFA	11 (10–14)	8 (4–13)	10 (5–14)
APACHE II	30 (28–36)	22 (17–28)	24 (16–28)

Values are presented as medians with quartiles.

## Data Availability

The data presented in this study are available on request from the corresponding authors.

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
