# Peer review of "Non-Neuronal Acetylcholinesterase Activity Shows Limited Utility for Early Detection of Sepsis"

_biomedicines, 2023, doi:10.3390/biomedicines11082111_

Round 1
Reviewer 1 Report
The authors address a very interesting topic.
The research on biomarkers in the course of sepsis is a very recent and at the same time much studied topic. The paper presented by the authors represents a continuation of previous studies (see references 6-14). However, the current topic has differences and biases.
Writing that the "conventional inflammation biomarkers" are CRP and WBCC is very simplistic (see line 83)
I invite the authors to expand on current research on other biomarkers of sepsis and organ failure in the "Discussion" section. I speak, specifically of High Mobility Group Box1 (HMGB-1), TGF beta1, Free Light Chains, Nephroceck. I invite the authors to deepen this theme and expand the references.
Author Response
We appreciate the reviewer's interest in other biomarkers of sepsis and organ failure. While our focus was on AChE activity and its relationship with disease progression and outcomes, we recognize the significance of exploring additional biomarkers in sepsis research. In the revised manuscript, we have included a brief discussion mentioning the importance of other biomarkers, such as High Mobility Group Box1 (HMGB-1), TGF beta1, Free Light Chains, tissue inhibitor of metalloproteinases-2 (TIMP-2) and insulin-like growth factor–binding protein 7 (IGFBP7) (Nephrocheck), as potential avenues for future investigation. We have also expanded the references to include relevant studies on these biomarkers to acknowledge their significance in sepsis research.
Furthermore, to address the concern about oversimplification, we have revised the statement referring to "conventional inflammation biomarkers" (line 83) to provide a more accurate description of these biomarkers and their role in sepsis.
Reviewer 2 Report
This work re-evaluated the diagnostic and prognostic roles in sepsis. The data is sufficient for the conclusion except the cohort is bit of too small.
In addition, the supplemental figures and tables can be moved to the main text.
In addition, more discussion shall be given to compare the methodology and patient cohorts to previous reports in order to justify the discripency.
Minor editing is required
Author Response
We appreciate the reviewer's observation regarding the sample size. We agree that a larger cohort would strengthen the statistical power of our findings and allow for a more robust analysis. In the revised manuscript, we have acknowledged this limitation and emphasized the need for future studies with larger cohorts to validate our results more effectively.
Regarding the supplemental figures and tables, we understand the concern about their placement. In response, we have moved supplemental information to the main text to provide a more comprehensive and streamlined presentation of our results.
In order to justify the discrepancy between our findings and previous reports, we have expanded the discussion section to include a detailed comparison of our methodology and patient cohorts with those of relevant previous studies. This comparison helps highlight the differences and biases that may have contributed to the contrasting results. Additionally, we emphasize the importance of conducting disease-specific investigations to avoid extrapolating findings from one pathological condition to another.
Reviewer 3 Report
The author’s finding does not add on any new finding or value to the involvement of AchE in sepsis. This finding is already well know in the field of sepsis. Also author did not gave a clear mechanism behind their finding.
Author Response
We would like to express our gratitude to the reviewer for taking the time to provide feedback on our work. As researchers, we appreciate the opportunity to engage in constructive discussions regarding our findings. To the best of our knowledge, there are currently no published data explicitly demonstrating the involvement of non-neuronal acetylcholinesterase in sepsis. If the reviewer is aware of any such studies, we would be grateful for the opportunity to discuss and consider them in the context of our findings. While the involvement of cholinergic system in sepsis has been extensively documented, the contribution of non-neuronal AChE remains largely unexplored in this context. We acknowledge the significance of serum cholinesterase, an enzyme with a structurally comparable origin to AChE, in both sterile and pathogen-induced inflammatory processes. However, the specific role of non-neuronal AChE in sepsis remains unexplored. Our study sheds light on this aspect, and despite reporting limited value for AChE activity in sepsis, we firmly believe that negative findings are equally important to enhance our understanding of the complex mechanisms governing the cholinergic response during inflammatory processes. Lastly, it's essential to note that our work represents a clinical observational study, and delving into the molecular pathomechanisms underlying our findings is beyond the scope of this study design. We hope that our contribution will encourage further research in this area to deepen our comprehension of the role of AChE in sepsis and related inflammatory conditions.
Round 2
Reviewer 1 Report
The authors have made the requested changes. The paper can be accepted for publication
Reviewer 3 Report
After the revision with better constructed discussions, i am happy to accept this article in its present format. I congratulate the authors for the publication.